# Who Is Happier in China? Exploring Determinant Factors Using Religion as a Moderator

**DOI:** 10.3390/ijerph16224308

**Published:** 2019-11-06

**Authors:** Yingying Sun, Yue Zhang

**Affiliations:** 1Institute for Disaster Management and Reconstruction, Sichuan University, Sichuan 610200, China; sunying@scu.edu.cn; 2School of Journalism, Sichuan University, Sichuan 610200, China

**Keywords:** happiness, religious identity, religious practice, health, politics, social relationships

## Abstract

The relationship between religion and happiness has been seriously understudied in non-Western and non-Islamic societies. Taking religious identity and religious practice as strata, the 2015 Chinese General Social Survey (CGSS) data were used to examine the predicting power of health, politics, and social relationships with regard to happiness in four different groups, as follows: People with a religious identity and practice, people with no religious identity but with a practice, people with a religious identity but no practice, and people with neither a religious identity nor practice. Multiple regression analyses were conducted using the Ordinary Least Squares method. The results demonstrate the influence of the independent variables in the four groups, thus confirming the expectation that different religious practices, as well as identities, play a vital role in moderating the degree of happiness. Physical and mental health are significant predictors of happiness regardless of different religious situations, with the effect of mental health here found to be greater in magnitude on happiness than that of physical health. Political participation was not found to be related to happiness, but having a left-wing political attitude did emerge as strongly predicting happiness. The results concerning social relationships further consolidate the hypothesis that religious practice should be taken into consideration separately from religious identity. This study indicates the importance of further investigating religious practice as an independent factor in religious studies in the context of Chinese society.

## 1. Introduction

Since the late 1980s, an increasing volume of empirical research has demonstrated the salutary effects of religious commitment on public health [1], life satisfaction [2], and personal feelings of happiness [3,4]. In these studies, religion has been treated as the independent variable [5,6], dependent variable [4,7], and mediator as well as moderator [8,9], which has helped to reveal some of the causes and consequences of religious involvement in highly topical social issues. Most literature, to date, has focused on societies dominated by the Judeo-Christian-Islamic tradition, particularly the United States, with the religion-happiness relationship being seriously understudied in non-Western and non-Islamic societies [10,11]. It was not until recently that several studies examined the correlations between happiness and religious commitment in Mainland China and Taiwan [11,12], whose societies have been dominated by Chinese traditions for thousands of years, such as Buddhism, Taoism, Confucianism, and folk beliefs [13]. However, while demonstrating the influence of religious identity, beliefs, and practices on happiness [11,12], none of these studies have investigated the determinant factors of happiness by considering the interaction between religious identity and practice. 

Regarding studies on people’s religious commitment in China, Lu and Gao [12] found that 22% of respondents identified themselves as religious, while 36% of respondents engaged in some form of religious practice across the various religions (e.g., folk religion, Buddhism, Confucianism). Lu et al. also reported that the majority of their respondents (33%) carried out practices that form part of Chinese folk religions, including various ritual practices such as ancestor worship, exorcism, and divination [12]. Most folk rituals are carried out at home or at the shrine of the particular locality and god [14], which might be one of the reasons why those previous studies found a greater number of people engaged in religious practices than those who defined themselves according to a religious identity. Hence, given the existence of a research gap in the differentiation between religious identity and practice, there is arguably value in investigating the influence on happiness as moderated by the interaction between religious identity and practice.

### 1.1. Religion and Religion Measuring 

Religion was the least studied social issue during China’s one-hundred-year process of social modernization [15]. However, since China adopted its policy of reform and opening up in the late 1970s, religion became a significant topic both in political discourse and in the academic community [16]. Today, it is widely recognized that Buddhism, Taoism, Protestantism, Islam, and Catholicism comprise the five major religions in China, alongside folk religions in rural areas [17,18,19]. Scholars have discussed the intrinsic characteristics of those specific religions from various perspectives, such as anthropology and philosophy [14]. Over the past 20 years, studies have adopted other perspectives, such as those drawn from sociology and politics, in order to enrich the field of religious studies [20]. Out of all of these topics, the social impacts of religion on contemporary Chinese society, especially its functions of enhancing social stability and harmony, have received the greatest attention [21,22]. Nevertheless, very few studies have quantitatively examined the impact of religion on Chinese society or differentiated between religious identity and practice on individual happiness. 

In the context of investigating the social function of religion, how to measure religion becomes a key issue. Multi-dimensional definitions and sophisticated indexes have been developed to enable this measurement in addition to that of religious identity. Among these, Glock [23] and, later, Stark [24] proposed a five-dimensional model by which to measure religiosity, including the aspects of religious beliefs, religious practice, religious feeling, religious knowledge, and religious effects. Glock pointed out that in studying religious beliefs, one may inquire simply into what people believe. For its part, religious practice encompasses both public practices, such as church membership or church attendance and the like, and private practices such as prayer and meditation. Glock emphasized that the first element of studying religious practice is that of paying attention to the frequency of engaging in ritualistic activities [23,24]. Public practice, as a collective activity, is believed to related to social relationships or social capital, as has been clearly observed in the Puritan, Catholic, and Jewish communities. Conversely, the private practices more frequently found among adherents to Buddhism, Taoism, and Chinese folk beliefs, have not received as much attention as public practices. The Glock–Stark approach has set a baseline for measuring religion. Although some critiques of this approach have emerged, subsequent research has still tended to follow the Glock–Stark model [25,26]. As Glock [23] pointed out, that past studies of religion have focused on assessing its broader expressions, thus raising the question of whether religiosity manifested in a particular way has anything to do with its being expressed in other ways. Recently, Chinese scholars have proposed a reconsideration of the measure of religion from the typically employed perspective of religious identity to using both religious identity and religious practice [27], particularly in social contexts where religious identity and religious practice do not align with one other.

Scholars have typically focused on employing religious identity to investigate religion-related issues in China, as most of the available research has done thus far. It is here argued that religion should be discussed in light of both religious identity and practice, given that ordinary religious believers in this context need not necessarily undergo admittance procedures or rites in order to acquire a religious identity. This is especially true for Buddhism, Taoism, Confucianism, and folk beliefs. On the other hand, the modalities of ‘doing religion’ are easy to acquire by, for example, simply following the local/ethnic tradition, or proclaiming allegiance to a certain religion in private [16]. Religious practices such as fortune-telling, Feng Shui, burning incense at home, and worshiping ancestors are easy-to-acquire, common religious practice that can be done in private. Although seemingly contradictory, it has empirically been proven that some Chinese people may not identify as having an explicit religious identity while still participating in religious activities. On the other hand, others may acknowledge their religious identity but lack a practice due to subjective or external reasons [28]. Therefore, the number of people who self-reported in the survey as having participated in religious practices is believed to be much bigger than the number of those claiming a specific religious identity. For instance, in the *Chinese General Social Survey 2012*, the number of folk religion believers was recorded as 3.5%; however, by asking people specifically about the folk belief practices they had participated in, this number surged to 52% [29]. Based on this, scholars inferred that the number of believers actively involved in organized religion was far lower than those in non-organized activities [27], which may have fortified the decision to measure religion simultaneously by identity as well as practice. 

In reviewing the extant literature, it becomes evident that happiness has been recognized as closely aligned with public health preparedness, or with enhancing health security [30,31]. While past studies have examined the direct causal relationship between religion and happiness, this has been done with the intention of exploring the effects of religious practice. Arguably, taking religion as the independent variable may not be an appropriate path for realizing this research goal since happiness here refers to a subjective feeling, while religious practices consist of various kinds of collective or individual human behaviors that do not directly influence subjective perception but through the moderation or mediation of other psychological factors. Other research on happiness has ascertained it to be moderated or mediated by feelings of autonomy and competence, whereby prosocial behaviors improve one’s sense of well-being [32,33,34]. In addition, evidence from social surveys and field work have revealed the frequency of participating in religious practices to be unexpectedly low. In certain surveys, more than one-third of Buddhist respondents in China were found to pay a visit to temples less than once a year and 57.4% were found to engage in meditation less than once a year [35], while in another authoritative social survey 0.04% of Christians admitted going to church or other religious places more than once a week and 0.26% admitted to praying more than once a week [36]. While the exact data results vary somewhat, they all reflect the fact that the frequency of Chinese people’s religious practice although vitally important, may not support taking religion as a major determinant of happiness in relevant research. 

As Frazier et al. [37] concluded, given that moderators address “for whom” or “when” a variable most strongly predicts an outcome variable, employing religious identity and religious practice as moderators would answer more specific questions about the influence of the proposed variables on happiness. Some previous studies have inspired the method of using religious items as moderators, especially in the domains of psychology and mental health [8,38]. Preliminary studies have demonstrated that the inclusion of religion related items could moderate certain predictors of psychological well-being [9,39], which, in turn, inspired the current research to employ religion as a moderator instead of an independent variable. This paper, therefore, aims to fill some of the gaps that have been identified in current studies on happiness by considering the interaction between religious identity and religious practice, taking religion not as a major factor of happiness but as a moderator that can influence other determinants of happiness. It is hoped that this approach may reveal more complex findings concerning religion and happiness, alongside the existing research that takes religion as the dependent or independent variable, which may thus open up a more in-depth understanding of religion. As Baron and Kenny discuss, a moderator functions to partition factors into subgroups that established the factors’ domains of maximal effectiveness with regard to a given dependent variable [40]. In the present research, religious identity and practice are applied as strata by which to divide the current survey samples into four groups, as follows: People with neither a religious identity nor practice (no RI × no RP), people with no religious identity but with a practice (no RI × RP), people with a religious identity but no practice (RI × no RP), and people with both a religious identity and practice (RI × RP). 

### 1.2. Health, Social Relationships, and Politics

To date, the previous literature has identified various influencing factors of happiness. The most important one can be seen as health related. Regarding health outcomes, studies have reported reciprocal relationships between happiness, mental health, and physical health [41]. Using a convenience volunteer sample of Muslims, Abdel-Khalek and Lester [42] found that people who see themselves as religious were more likely to have greater levels of mental health and happiness. Studies extensively suggest religion to be linked with lowered depression, improved mental and physical health, and reduced mortality [43]. For instance, the findings of Strawbridge et al. [44] indicate that frequent religious attendance reduces the mortality risk among community-dwelling residents, partly due to enhanced social ties and improved health behaviors. Respondents reporting life-changing religious or spiritual experiences have been found to have fewer depressive disorders and fewer anxiety disorders and symptoms [10]. Nonetheless, no study has yet examined the health impact on happiness by differentiating between religious identity and practice. The present study attempts to address these gaps by answering the following questions:

RQ1: Does having a religious identity or religious practice present moderating effects on physical or mental health?

RQ2: Does mental health exert more influence over happiness than physical health under the moderating effect of religion?

Another important factor influencing happiness is that of social relationships. The latter can be regarded as a set of people, or groups of people, interacting with each other [45]. Scholars have found evidence of a link between the strength of social relationships and increased levels of happiness [2,43]. Research on the effects of religion on happiness has found that church attendance and the close social support occurring within the church community are the most important factors, with many reporting their closest friends also to be church members [46]. Given the relative social isolation and transportation difficulties of many elderly people, religious congregations that reach out to their members may be seen as providing critically important sources of friendship and support [47]. As such, the findings of previous studies indicate that religious practice has the function of enhancing the relationship between social relationship and happiness. Building on this, the current study attempts to address the following question:

RQ3: Does having a religious practice influence the correlation between social relationships and happiness?

Political participation has been examined in diverse social contexts in order to uncover its connection with happiness, as such participation has been considered to be the inheritance or continuation of Aristotle’s, as well as other classical philosophers’ thinking; namely, that participation in political matters is an essential part of individual fulfillment and, therefore, individual happiness [48]. A cross-European analysis asserted that political participation displayed a robust positive impact on life satisfaction [49]. Frey and Stutzer [50] proved that the more developed the institutions of direct democracy in which voters had the final say, the happier the individuals were. Wallace and Pichler [51] developed the perspective that by taking part in social or political participation activities, individuals could realize their self-actualization and could, therefore, achieve higher levels of wellbeing. Even though there exists a long-established tradition of studying the relationship between political participation and happiness, the relationship between the two has received comparatively little empirical attention in China. When intertwined with religion, this attention has been even scarcer. Regardless of different social contexts, voting is one of the most important and common activities within the remit of political participation [52]. Given that grassroots-level elections comprise the basic and most important form of political participation in both urban and rural China [53,54], the current research takes grassroots election voting behavior as an indicator by which to measure political participation in China, posing the following question:

RQ4: Does religious identity or religious practice present moderating effects on political participation?

Regarding political orientation, prior research has demonstrated that political conservatism is associated with increased happiness [55] as found, for example, by Napier and Jost [56]. Using data from the *2012 General Social Survey* and the *2005 World Values Survey*, Bixter [55] concluded that religiosity had a greater effect on happiness among more politically conservative individuals than those identifying themselves as more politically liberal. As China has no such thing as conservative or liberal political parties, scholars have applied various political factors to investigate the political influence on happiness in the country. For instance, using the *2012 East Asian Social Survey*, Fu [57] indicated that people’s political attitudes or behaviors may exert an influence on happiness. However, very few studies have directly examined whether political attitudes and political participation can predict happiness in China. To respond to this research gap, the present study adopts the populous definitions of left- and right-wing political ideologies in order to explore their related impacts on happiness. For the purposes of this paper, left-wing ideology in China is taken as indicating people who support “state power in politics”, while, conversely, right-wing ideology refers to “individual rights and freedom”. It should here be noted that, in the context of Chinese political and social commentary, the right-wing is aligned with the liberal ideology and the left-wing with the conservative. Hence, the following question is put forward:

RQ5: Do individuals identifying as politically left-wing tend to feel happier than those identifying as right-wing under the moderating effect of religion?

### 1.3. Demographic Factors

Other influencing factors on happiness include demographic characteristics such as age, gender, ethnicity, education, marriage, and income. Age effects represent aging-related developmental changes over the life course, which have been taken and developed into a methodology of age-period-cohort models to predict individuals’ happiness [58]. Thus far, the age effects of happiness are debatable, as previous studies have theorized two possibilities regarding the effect of age on happiness, as follows: The U-shaped curve theory and the age-reducing theory [59,60]. Although studies report the age effects of happiness in China to be U-shaped [60,61], the moderating effects of religion on the relationship between age and happiness remain unclear. Regarding gender, Koenig, King, and Carson [10] reveal that women are more likely to attend religious services, pray privately, say religion is important in their lives, and depend on religion as a coping behavior. They state that it is possible that religious beliefs and practices are more deeply ingrained in the social and psychological lives of women and, therefore, confer greater health benefits upon them. With regard to education, Hartog and Oosterbeek [62] argue that education correlates strongly and positively with happiness scores in poor nations, and weakly in rich nations. In terms of income, people in the wealthiest nations tend to be leaving organized religion, or to have no specific religious affiliation [7]. For example, Koenig, King, and Carson [10] noted that the vast majority of Scandinavians are atheists or non-religious; at the same time, Scandinavian nations tend to experience the highest levels of subjective well-being in the world [7]. Regarding marriage, Myers [63] reports that religious faith and marriage seem to be good predictors of happiness. Similarly, Ellison, Gay, and Glass [2] found that religious beliefs and practices tend to promote psychological well-being and social support among partners, all of which have been linked to better marriages. In the present study, age, gender, ethnicity, education, marriage, and income were included as control variables for the analysis models. 

The results of this research could provide academic insights into how religion moderates the determinant factors of the happiness of Chinese people. The theoretical framework of this paper is shown in Figure 1.

## 2. Materials and Methods 

### 2.1. Data Collection

The data used in this research were drawn from the *2015 Chinese General Social Survey* (CGSS), a series that has been conducted annually since 2003. The 2015 CGSS questionnaire contains the following six modules: Core module, ten-year retrospective module, the East Asia Social Survey, the International Social Survey, energy module, and legislation module. For the purposes of the present analysis models, the variables were selected from the core and ten-year retrospective modules. The questionnaires of these two modules comprise the constant parts of the CGSS series. The 2015 CGSS data were collected in 2015 by the National Survey Research Center of the Renmin University of China. A stratified multi-stage sampling method was used, with a face-to-face interview method. A total of 10,968 respondents were engaged. The samples with missing values for the selected variables were disregarded in the analysis, meaning that a total of 9699 cases were ultimately included in the models. 

### 2.2. Measurements

Respondents’ happiness was taken as a dependent variable in this research. The relevant question here was as follows: “In general, do you feel happy?” The answer was measured by a Likert scale ranging from 1 to 5, respectively indicating “very unhappy” to “very happy”. 

“Religious identity (RI)” and “religious practice (RP)” were used as strata to divide the samples into four groups. Respondents were asked to report their exact religious identity, with those reporting no religion coded as “0”. As long as the respondent reported having a religion (Buddhism, Taoism, folk religion, Islam, Catholicism, Christian, etc.), their answer was coded as “1,” regardless of the exact religious identity. The variable of religious practice was measured by asking “how often do you participate in religious activities?” Respondents were required to choose one answer from the following options: Never, less than once a year, 1–2 times a year, several times a year, once a month, 1–2 times a month, almost once a week, several times a week. The answer “never” was considered as representing “no religious practice”, while the other answers represented that the respondents “have a religious practice”. The reason for recoding the original 7-scale religious practice into a dummy variable has been explained in the current literature review. Namely, former studies have revealed a low participation frequency of religious practice; thus, the binary coding is believed to be adequate for the purpose of analyzing religious commitment in practice.

Physical and mental health, social relationships, and political attitudes and participation were the main explanatory variables applied in the current research. Respondents were asked to use five-point scales to self-estimate the condition of their physical health, from “very unhealthy” to “very healthy”, as well as their mental health condition. Political participation was measured by asking the following: “did you vote in the last grassroots-level election?” The three response choices were “yes”, coded as 1, “no” and “not qualified”, the latter both coded as 0. Political attitudes were measured by three statements regarding the person’s basic attitude towards the government, as follows: “the government should not intervene if someone criticizes it in public”; “giving birth to a child is a personal matter and the government should not intervene”; “it is the people’s freedom to choose where to live and work, and the government should not control this.” The answers ranged from “strongly disagree” (coded as 1) to “strongly agree” (coded as 5). The more the respondents agreed with the statements, the more right-wing they were deemed to be. In the analysis, the three variables were computed into a single variable in the regression equation. The final explanatory variable was that of “social relationships”, measured by asking the following: “what is the degree of intimacy between you and the people around you?” These responses also adopted a five-point scale, from “very unfamiliar” (coded as 1) to “very familiar” (coded as 5).

Six other demographic variables were introduced in order to examine the influences of demographic features. Respondents were asked to report their gender (with male coded as 0 and female as 1) and age in the year 2015. Respondents were also asked to report their ethnicity, with “Han people” coded as 1 and “minority groups” as 0. In order to explore the support-giving function of religion in unhappy marriages, this research established the dummy variable of “marital status”, in which “divorced or widowed” respondents were coded as 1 and those who had had no such experiences (regardless of whether they had ever been married or not) were coded as 0. Regarding answers to “education background”, 1 to 5 represented a five-stage scale from “illiterate” to “primary school or lower”, “middle school”, “high school”, and finally “under-graduate and above”, in sequence. In addition, respondents were asked to report their personal annual income in 2014 in CNY (Chinese Yuan).

### 2.3. Data Analysis Strategy

This research examined the moderating effects of religious identity and practice on happiness by using religious identity (RI) and religious practice (RP) to divide the respondents into four groups, as follows: no RI × no RP, RI × no RP, no RI × RP, RI × RP. Multiple regression analyses of the Ordinary Least Squares method was employed to test the predictive power of the independent variables. In order to verify and compare the predictive power of the independent variables, coefficients between the independent variables and happiness in each regression analyses of the four groups were standardized and the Standard β (std. β) obtained, so as to directly display the predictive power of key determinants under the moderating effect of religious identity and religious practice. The analyses were processed using EViews 10, IHS Global Inc. (Irvine, CA, USA).

## 3. Results

A total of 9699 cases were used for the analyses. This section presents the results of the descriptive analysis and regression analyses for the four groups.

### 3.1. Results of Descriptive Analysis

The descriptive analysis presented a general picture of all variables used in this study (Table 1). The distribution of happiness (M = 3.87) showed that the respondents were quite optimistic about their lives, as 78.4% of them reported generally feeling happy, of whom 17.7% gave “very happy” as their answer. A total of 89.7% of respondents claimed they had no religious identity, while 10.3% reported having one, including those based on Buddhism, Taoism, folk beliefs, Protestantism, Catholicism, Islam, and so on. Yet, 8394 respondents (86.5%) reported that they had never participated in any religious practices, while 1305 respondents (13.5%) claimed that they had participated in religious activities irrespective of the frequency of the religious participations. 

A total of 47.0% of the respondents were male and 92.7% identified themselves as being of Han ethnicity, with 7.3% being from minority groups. The age of the whole sample ranged from 18–94 (M = 50.5), with 50 years as the median, 38 as the younger quarter, and 63 comprising the last 25%. The distribution of education displayed a tendency towards a lower level, with 36.9% of the respondents reporting only having undergone primary school education, and 28.7% middle school. Only 16.4% of the total respondents had had a higher education experience. In terms of personal annul income, 31,452.31 CNY was the mean value, 3000 marked the line of the lowest quarter, and 36,000 the highest quarter.

Respondents’ self-reported physical health (M = 3.62) and mental health (M = 3.84) conditions emerged as relatively positive in this survey. A total of 38.8% of the respondents claimed that they were physically healthy, with 21.8% reporting being “very healthy”; specifically, 42.8% reported they were healthy in terms of their mental health, with 25.3% of the latter recording “very healthy”. With regard to political participation, 47.3% of the respondents responded “yes” to partaking in this, while the majority, 52.7%, stated that they did not participate. In terms of political attitudes (M = 3.05), overall the respondents showed a moderate liberal stance. The mean value of social relationships was found to be 3.74, with only 2.4% and 12.3% of respondents respectively claiming to be “very unfamiliar” or “unfamiliar” with the people around them, and over half of the respondents stating that they were “familiar” (33.3%, *n* = 3227) or “very familiar” (28.8%, *n* = 2796) with the same.

### 3.2. Religious Identity and Practices as Moderators in Exploring Determinants of Happiness

The purpose of this study was to explore the determinants of happiness under the moderating effect of religious identity and religious practice. Table 2 presents the results of the multiple regression analyses across the four groups (no RI × no RP, RI × RP, RI × no RP, no RI × RP) that examined the predictive power of the independent variables. 

Firstly, in group “no RI × no RP” (*n* = 8172, 84.3%), physical health (std. β = 0.122, *p* < 0.001) and mental health (std. β = 0.234, *p* < 0.001) initially emerged as significantly related to respondents’ sense of happiness. Political attitudes were found to be closely related to happiness; specifically, those who identified with the left-wing ideology in China reported feeling more happiness (std.β = −0.017, *p* < 0.001) than those identifying as right-wing. Social relationships (std. β = 0.050, *p* < 0.001) and age (std. β = 0.090, *p* < 0.001) were also strong indicators of happiness, although the co-efficient values of the two were relatively low. Marital status also yielded happiness (std. β = −0.040, *p* < 0. 001), as none of the divorced or widowed respondents were happier than those who reported not having had these experiences. Personal annual income was found to be significantly related to happiness (std. β = 0.281, *p* < 0.1). These findings of the main determinants predicting happiness in secularized China, without the moderation of either a religious identity or practice, are worthy of reflection. This is so since, according to different statistical data such as the *China Family Panel Studies*, while the specific percentage fluctuates, most Chinese respondents claimed to have neither a religious identity nor practice.

With regard to group “RI × RP” (*n* = 775, 8.0% ), physical (std. β = 0.144, *p* < 0.001) and mental (std. β = 0.215, *p* < 0.001) health were found to exert a pronounced effect on happiness, as well as to be the two most influential factors determining the latter, according to the standardized coefficient value. Even with the comfort of religion, respondents who were divorced or widowed (std. β = −0.040, *p* < 0.1) were found to be less inclined towards happiness than those who were not. In addition, personal annul income (std. β = 0.294, *p* < 0.1) and age (std. β = 0.102, *p* < 0.01) also demonstrated a significant predictive power over happiness, with similar coefficients as in the “no RI × no RP” group. This indicates that with or without the moderating effect of religion, people of an older age or a higher personal annual income were more likely to feel happiness. However, under the moderating effect of religion, the formerly significant determinants of political attitude and social relationships were not related to happiness.

In group “RI × no RP” (*n* = 222, 2.3%), the respondents were identified as having a religious identity but no practices. Age was, again, found to be positively related to happiness (std. β = 0.134, *p* < 0.1). The analyzed data revealed that having healthier physical (std. β = 0.140, *p* < 0.01) or mental health (std. β = 0.164, *p* < 0.01) was positively related to happiness. More intimate social relationships were found to lead to a greater sense of happiness, as affirmed by the regression results (std. β = 0.125, *p* < 0.01). Political attitudes were found be related to happiness under the moderating effect of religious identity, with those identifying as left-wing happier than those who were right-wing (std. β = −0.042, *p* < 0.1). Since most of the previous quantitative research concerning religion and happiness has only considered religious identity, the results from the “RI × no RP” group could be referred to by way of a cross-cultural comparison. 

In the column that reveals the regression results of group “no RI × RP” (*n =* 530, 5.5%) in Table 2, comprising respondents who claimed having a religious practice but no religious identity, provide a unique finding for the current research. According to the standardized coefficient value, mental (std. β = 0.133, *p* < 0.001) and physical (std. β = 0.126, *p* < 0.001) health emerged as the two key determinants of happiness that weighed much more than the other factors. However, the probability of social relationships did not show significance, indicating that there is no relationship between this factor and happiness for those who do have a religious practice but no religious identity. A new finding emerging from this group in comparison to the other three is that ethnicity was found to indicate happiness. Specifically, those who identified as Han people (std. β = 0.067, *p* < 0.1), with no religious identity but engaging in religious practices, were found to be happier than people from minority groups with the same religious characteristics. In addition, none of the other independent variables under discussion in the present study were related to happiness in this group.

Overall, mental and physical health emerged as the two most influential factors in determining happiness. According to the variance of the standardized coefficient value among the four groups, RQ1 could be answered to the effect that religious identity, as well as religious practice, presented moderating effects on physical and mental health. Religion related variables were found to moderate the ecoefficiency of the two factors, with mental health consistently subjected to these moderators more prominently, so an affirmative answer was derived for RQ2. Social relationships were proven only to be moderated by religious practices in association with happiness; this, rather than enhancing the relationship between social relationships and happiness as queried by RQ3, meant that religious practice moderates the relationship between social relationships and happiness by rendering the former insignificant. The same case emerged with regard to political attitudes, only without a religious practice, whereby, regardless of religious identity, people with a left-wing ideology were found to be happier than those with a right-wing one, thus answering RQ5. Political participation was found to have no significance among the four groups, thus answering RQ4 to the effect that neither religious identity nor practice presented with moderating effects on political participation. In addition, age, ethnicity, marital status, and personal annual income were demonstrated to be influenced by the moderators; however, the power of political participation, education, and gender over happiness remained insignificant regardless of the moderating effect of religion.

## 4. Discussion

The present study contributes to an in-depth understanding of the predictive power of health, politics and social relationships over happiness under the moderating effect of religious identity and religious practice, based on a large volume of canonic research. The findings of this quantitative research contribute to the exploration of the influence of religion in China, opening up a new perspective by taking religious identity and religious practice separately into consideration, and, innovatively, uncovering the influence of religious practice as a moderator in itself. 

It should be emphasized that this research consolidates previous propositions positing that religious practice differs from religious identity [64], and should thus be inspected as an independent variable in addition to religious identity. When religious identity was controlled for in the present study, engaging in religious practices or not led to different associations and vice versa. That is to say, religious identity, on its own, was not found to be sufficient in building a complete picture —religious practice is also of great importance. While distinguishing characteristics have been mentioned by a few previous scholars [65,66], the current research can, to some extent, be seen as pioneering in terms of its exploration of the social influences of religious practices. Overall, the first major contribution made by the current research is the insight that, in order to analyze religious issues in China in the future, religious practices should be taken into consideration as a vital factor.

The first prominent finding meriting discussion is that both good physical and mental health were found to have a significant positive effect on happiness under the moderating effect of religious identity and practice. Although physical and mental health were both found to predict positively happiness in the four designated groups with different religious identity/practice combinations, the comparable standardized coefficient values of both types of health varied across the different groups. For instance, comparing “RI × RP” and “no RI × RP” with religious practice controlled for, people who reported having a religious identity were found to be much happier than those who did not have a religious identity when the health of their physical or mental state increased. In addition, it is worth noting that across the four sampled groups, an increase in mental health consistently lead to a higher degree of happiness, although both physical and mental health demonstrated slight fluctuations in coefficients. Under certain conditions, such as among those in groups “RI × RP” and “no RI × no RP”, the standardized coefficient values of mental health were nearly twice those of physical health. This may indicate that the condition of one’s mental health is a greater determinant of people’s happiness than previous studies have implied. The reason for this might be that the sense of gratitude associated with happiness—a mental rather than physical attribute—is considered a vital item on indices of psychological distress [11].

Secondly, the findings of the current research uncovered a frequently used variable affecting happiness, social relationships, on which religious practice was revealed to bear a strong moderating effect. For those who did not participate in religious activities, whether they had a religious identity or not, social relationships were found to help increase their happiness. These results may shed new light on previous findings that religion could work as a substitution for social relationships and, as such, enhance a sense of happiness [67]. The current research further discovered that it is not religious identity but, rather, religious practice that exerts a substitution effect in terms of social relationships. This builds on previous research, such as that of Durkheim [68], which emphasized religion’s social function as benefiting mental health. Likewise, Bartkowski et al. [67], grounded in Durkheim’s approach, found that only communal prayer can yield anxiety-reducing benefits. The present research takes a step further in uncovering that religious practice shares certain intrinsic similarities with social relationships, on account of the people involved seeking psychological consolation. When conceptualizing religious practice, this research also highlighted its duality. With regard to public practice, this research may demonstrate that collective religious activities acted as a substitute for people’s needs in terms of social connection; in terms of private practice, intrinsic individual cultivation fulfilled their needs in terms of social connection. Moreover, all of the above combine to prove, once again, that religious practice is a vital variable to consider in religion-related research, and is also worth considering in studies concerning people’s sense of well-being and broader social disciplines.

Regarding political participation, while this was expected to relate to happiness, in none of the four study groups did the results show any relating association between political participation and happiness. This finding may seem confusing in a Western context, where religion and political activism have historically been intertwined [55,56,69,70]. Conversely, in China—one of the most commonly acknowledged secular countries in the world, especially in contemporary times – distance is encouraged between religion and politics [71], from which avoidance is in accordance with the doctrines of Buddhism and Taoism. It is worth noting here that respondents who neither identified as having a religious identity nor religious practices still did not relate their happiness to political participation. This may fortify the assertion that political participation does not affect people’s sense of happiness [55]. The depoliticization across the entire Chinese social domain that took place as a result of the country’s cultural revolution could be one explanation for this non-connection between political participation and happiness [72]. A further finding here was that political attitude emerged as a relatively active determinant in this research and was prominently subjected to the moderating effect of religious identity. Political attitudes were found significantly to affect happiness in groups “RI × no RP” and “no RI × no RP”, implying that when not engaging in any religious practices, believing in a certain religion or not will affect the predictive power of one’s political attitude. The people who reported having a more left-wing ideology in these groups indicated higher levels of happiness, which may be explained by what Zhang and Sun [73] discussed; namely, that the Communist Party of China and the Chinese government venerate a left-wing ideology, leading to the implication that people who have the same political attitude as the ruling administrative bodies would be happier than those who do not. However, further research on this is required in order to explain why only religious identity was found to moderate political attitudes. 

Regarding the demographic factors examined here, age, personal annual income, and marital status were found to be the three most significant in terms of predicting happiness. Apparently, more senior people were more likely to report a higher sense of happiness than younger ones, which neither consolidated the U-shaped curve theory nor the age-reducing theory mentioned in the literature review [59]. This increase of happiness with age may be due to elderly people’s relief from family support, as their children have grown up to support themselves, and their parents have passed away and no longer incur any expenditure. Our results showed that significant positive effects of personal annual income on happiness were only found in two groups, “no RI × no RP” and “RI × RP”. It is arguably easy to understand that the former groups, who are totally secular people, become happier when earning more money. However, people who reported having a religious identity and practice were found to be deeply influenced by religious doctrines such as “not being pleased by external gains, not saddened by personal losses”, and “staying aloof from the mortal life”, thus making it somewhat confusing that they reported having the same feeling about earning money and happiness as the former group. Nevertheless, as a fact, most Chinese people, especially businessmen have beliefs in the God of Wealth, and make enshrine and worship (e.g., incense, wine, meat, chicken, fruits) periodically, which may be the reason why income and happiness were found to be significantly correlated in the group of “RI × RP”. Marital status emerged as closely related to those who had both a religious identity and engaged in religious practices. This finding may indicate that, together with the moderating effect of religion, unhappy marriages only had limited influence over respondents’ general happiness. 

Regarding ethnicity, the moderating effect of religious practice was relatively more marked than that of religious identity. For all those who reported participating in religious practices, only Han people without a religious identity were found to be statistically happier than respondents from minority groups. While this may, superficially, seem hard to understand, this finding conforms to the current Chinese social context, in that people from minority groups are more inclined to recognize their religious identity from their earliest life stages [14]. Thus, having a religious practice alone would not, in itself, lead to a sense of happiness among minority groups. In contrast, most of the Han respondents did not report having any religious identity, notwithstanding visiting monasteries or temples, which is arguably a behavior that could simply be comprehended as praying for fortune [14]. Thus, having no religious identity and only engaging in religious practices emerged as acceptable for the Han respondents in this study, and was even seen to bring them happiness. For education, it was not found to be related to happiness in any of the four groups, which was resonant with some highly-cited studies [74,75]. As for gender, numerous studies have reported no gender differences in happiness across countries [76], which the current research echoes. In this regard, it may be that Seligman’s explanation is well suited to Chinese society, namely, a male-dominated cultural context, where “women are both happier and sadder than men” [77].

The main limitation of the present study was that some of the key variables, like physical and mental health, were not composed of multi-level items to form an index, and may thus have weakened the reliability and validity of the results. This may be seen as an inevitable disadvantage of utilizing second-hand, especially general survey data in trying to conduct exploratory studies. Notwithstanding this, the research has preliminarily proved that religious identity and religious practice play a significant role in moderating the predictive power of factors like health, political participation, and social relationships over happiness. Applying a specific questionnaire survey with an in-depth index of determinants could thus be a valuable component of future research on happiness. 

## 5. Conclusions

The current study examines the determinant factors of Chinese people’s happiness by using religious identity and religious practices as moderators. This research may be seen as pioneering in terms of its adoption of religious identity and religious practice as two separate factors, with the results across the four constructed groups validating the necessity of considering religious practice as an independent variable. Physical and mental health were proven to positively influence happiness, which further supports the fact that, although occurring under different social contexts, these two variables strongly predict people’s sense of happiness. Furthermore, mental health was proven to be more influential on happiness than physical health. Although political participation was not found to be related to happiness, having left-wing political attitudes displayed a strong determinant effect over happiness, in accordance with China’s social context. The current research has thus proposed the new perspective of applying specific religious factors as moderators in the investigation of religion-related issues. This approach does not aim to decease the importance of religion but, rather, to expand the understanding of how religion exerts its effects in a secularized society. It is recommended that future studies continue to scrutinize happiness in explicit religious believer groups, and concentrate on the influences of specific health problems on happiness instead of discussing health as a general condition. 

## Figures and Tables

**Figure 1 ijerph-16-04308-f001:**
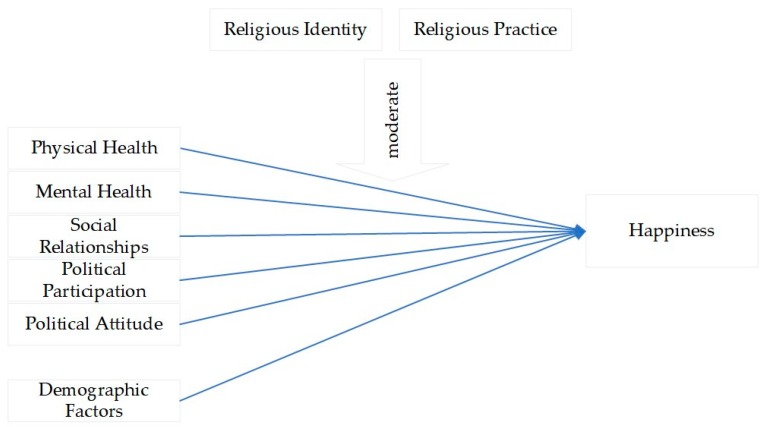
Research framework for the determinant factors of happiness as moderated by religious identity and practice.

**Table 1 ijerph-16-04308-t001:** Results of Descriptive Analysis (*N* = 9699).

	Frequency (*n*)	Percent (%)	Mean	Min	Max
**Happiness**			3.87	1	5
**Gender**			0.53	0	1
Male	4560	47.0			
Female	5139	53.0			
**Age**			50.5	18	94
**Ethnicity**			0.93	0	1
Han people	8988	92.7			
Minority groups	711	7.3			
**Marital Status**			0.11		
Divorced or widowed	1082	11.2			
No experience of being divorced or widowed	8617	88.8			
**Religious Identity**			0.10	0	1
Yes	997	10.3			
No	8702	89.7			
**Religious Practice**			0.13	0	1
Yes	1305	13.5			
No	8394	86.5			
**Education**			3.01	1	5
**Personal Annual Income**			31,452.31	0	9,991,500
**Physical Health**			3.62	1	5
**Mental Health**			3.84	1	5
**Political Participation**			0.47	0	1
Yes	4583	47.3			
No	5116	52.7			
**Political Attitude**			3.05	1	5
**Social Relationships**			3.74	1	5

**Table 2 ijerph-16-04308-t002:** Regression results of four religion moderating groups.

	No RI × no RP (*n* = 8172)		RI × RP (*n* = 775)		RI × No RP (*n* = 222)		No RI × RP (*n* = 530)	
	Coefficient	Standard β	Coefficient	Standard β	Coefficient	Standard β	Coefficient	Standard β
*Constant*	2.214	0.416	2.215	0.424	2.590	0.516	2.697	0.487
Gender	0.010	0.002	0.027	0.007	−0.051	−0.013	0.034	0.008
Age	0.005 ***	0.090 ***	0.005 **	0.102 **	0.008 *	0.134 *	0.003	0.045
Ethnicity	0.018	0.004	0.022	0.005	−0.108	−0.027	0.269 *	0.067 *
Marital Status	−0.160 ***	−0.040 ***	−0.160 *	−0.040 *	−0.206	−0.052	−0.125	−0.031
Education	0.008	0.008	0.026	0.026	0.022	0.022	−0.020	−0.02
Personal Annual Income	1.13 × 10^−7^ *	0.281 *	5.88 × 10^−7^ *	0.294 *	3.84 × 10^−7^	0.089	4.63 × 10^−7^	0.058
Physical Health	0.122 ***	0.122 ***	0.144 ***	0.144 ***	0.140 **	0.140 **	0.126 ***	0.126 ***
Mental Health	0.234 ***	0.234 ***	0.215 ***	0.215 ***	0.164 **	0.164 **	0.133 ***	0.133 ***
Political Participation	< 0.000	< 0.000	−0.011	−0.003	−0.115	−0.029	−0.045	−0.011
Political Attitude	−0.017 ***	−0.017 ***	−0.014	−0.014	−0.042 *	−0.042 *	−0.017	−0.017
Social Relationship	0.050 ***	0.050 ***	0.042	0.042	0.125 **	0.125 **	0.019	0.019
Adjusted R^2^	0.135		0.144		0.140		0.075	
F-statistic	116.863		12.856		4.267		4.884	

* *p* < 0.1, ** *p* < 0.01, *** *p* < 0.001.

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
