# Peer review of "Who Is Happier in China? Exploring Determinant Factors Using Religion as a Moderator"

_ijerph, 2019, doi:10.3390/ijerph16224308_

Round 1

Reviewer 1 Report

I reread the paper submitted by the authors and I do not see significant improvement or a serious attempt to confront the problems I raised in my review. For example, to my comment "The authors depended almost totally on self-report regarding religious identity and religious practice. These key variables need to be more specifically investigated by using questionnaires as well as interviews to ascertain more exactly the level of religious identity as well as the level of religious practice", the authors responded "As we used a secondhand data set, we have to respect the information that the original questionnaire investigators intending to collect from respondents". Thus the authors feel that despite the flaw in the original research they are examining, they continued to analyze the flawed data which is a very questionable practice.

In addition, in response to my comment "I would suggest that statistical procedures such as path analysis and structural equation modelling (SEM) are vastly superior to the analyses used by the authors as SEM can much more accurately pinpoint to the relationships between the research variables than the regression analyses used in the study", the authors responded "We have tried path analysis and structural equation modelling (SEM) to analyze our data. However, results were too poor to help us answer the research questions and hypothesis". Thus, because the robust path analysis and SEM analysis yielded poor results, the authors resorted to another less suitable statistical procedure to obtain better results. This is not acceptable at all. One cannot "go shopping" for alternative statistical procedures to procure better results when the recommended robust proceduresd do not produce the expected results The results presented in Tables 2-5 do not make up for the inherent weaknesses of the regression analyses. It would have been much more acceptable to conduct the suggested path analysis and SEM analysis and then make an honest attempt to explain the poor results. This would in no way would have detracted from the worth of the current study. 

Author Response

Reviewer 1

Comment 1: I reread the paper submitted by the authors and I do not see significant improvement or a serious attempt to confront the problems I raised in my review. For example, to my comment "The authors depended almost totally on self-report regarding religious identity and religious practice. These key variables need to be more specifically investigated by using questionnaires as well as interviews to ascertain more exactly the level of religious identity as well as the level of religious practice", the authors responded "As we used a secondhand data set, we have to respect the information that the original questionnaire investigators intending to collect from respondents". Thus the authors feel that despite the flaw in the original research they are examining, they continued to analyze the flawed data which is a very questionable practice.

Authors’ response: Thanks for the reviewer’s question. It is true that the series of Chinese General Social Survey (CGSS, 2003-2015) were conducted through face-to-face interview and all the questionnaires were responded according to responses’ self-report answers. This kind of investigation method was developed and highly acknowledged by other major social surveys such as Japanese General Social Survey, Taiwan General Social Survey, and East Asia Social Survey (http://www.issp.org/about-issp/). For example, in the 2012 JGSS Self-Administered Questionnaire A, two questions regarding religious identity and religious practice were given as follow: Q68-1 Do you follow a religion? Q68-3 How would you describe yourself as a religious follower? (http://www.issp.org/about-issp/) Hence, firstly, for the quality of the original data, we would like to believe that the 2015 CGSS do meet a widely acknowledged quality standard.

Regarding the self-report answers to religious identity and practice, it is true that using mixed method such as questionnaires, face-to-face interview, long-term field work, and action research could enable us to collect more accurate data. However, in that way, it is unavoidable that we, as investigators, can exert a certain of influence on research targets, once we start to talk to them (Sugiman, 2006). One of the advantages of using the secondhand data is that, the original investigators did not intentionally to get benefit on collecting information regarding religious issues. The main objective of the 2015 CGSS is to collect data on the general life issues of Chinese people, not specifically religion. Hence, secondly, for the self-report answers to religious issues, we would like to believe that the 2015 CGSS has exerted a minimum level on respondents, and the related data has an accepted quality standard.

In addition, in religious studies, plenty of previous researches have applied self-report answers regarding religion. Such research can be seen in Abdel-Khalek (2014), Abdel-Khalek and Lester (2017), Baroun (2006), Lu and Gao (2017), and so on. Hence, we would like to believe that using self-report answers to religious issues to examine the relationship between religion and happiness is acceptable.

Last but not least, this manuscript is our first step toward examining the relationship between religion and happiness in the context of Chinese society. Based on the reviewer’s suggestion, we are planning to apply mixed method to investigate and collect more concrete data on those two issues in ethnic communities in Sichuan Basin.

Comment 2: In response to my comment "I would suggest that statistical procedures such as path analysis and structural equation modelling (SEM) are vastly superior to the analyses used by the authors as SEM can much more accurately pinpoint to the relationships between the research variables than the regression analyses used in the study", the authors responded "We have tried path analysis and structural equation modelling (SEM) to analyze our data. However, results were too poor to help us answer the research questions and hypothesis". Thus, because the robust path analysis and SEM analysis yielded poor results, the authors resorted to another less suitable statistical procedure to obtain better results. This is not acceptable at all. One cannot "go shopping" for alternative statistical procedures to procure better results when the recommended robust procedures do not produce the expected results The results presented in Tables 2-5 do not make up for the inherent weaknesses of the regression analyses. It would have been much more acceptable to conduct the suggested path analysis and SEM analysis and then make an honest attempt to explain the poor results. This would in no way would have detracted from the worth of the current study.

Authors’ response: Thanks for the reviewer’s suggestions and comments. Actually, in order to find the best way to answer the research questions and hypotheses we have tried SEM analysis and other method to find the appropriate one. However, here are some reasons that we use the current method: 1) The authors of this paper could not design the questionnaire according to the research purpose but to use the second-hand data, so some of the observable variables we have to use do not have latent variables which would lead to a partially over-saturated issue, and would further disable parameter estimation. 2) We then tried both the OLS and Threshold Regression, and the results were consistent, which we believe demonstrate the robustness of chosen method. 3) The Threshold Regression could present the threshold values concerning the variation of age, which would help observe the age diversities of the determinants.

    We admit that SEM analysis is a very robustness method in social sciences, but it will under-identify the parameters and lead to model misspecification in the current research. So we choose an more appealable method to fulfill the research purpose.   

Reviewer 2 Report

.

Author Response

We have the language checked. Thank you!

Reviewer 3 Report

1.This article focuses on religion as a moderator between determinant factors of happiness and happiness.  But I need some more argumentation why religion is treated as moderator and not as independent variable. There is no substantion for this choice from the literature. The argumentation for choosing religion as a moderator points at: a) there are mediators between religion and hapiness en b) the frequency of participating in religious practices is low. Why is this a reason for choosing for moderation?

2.If one divides the moderating variable in 4 groups on the basis of religious participating and identity one expects an argumentation about the differences in these 4 groups regarding the connection between the determinants factors of happiness and happiness. But all the hypotheses are the same for the different groups. The hypotheses do no regard the moderating groups but the determinant factors.

There are no hypotheses for group 4: no RI * no RP. RQ2 (is mental health a greater determinant of people’s happiness than physical health?) deviates from the purpose of the article. In addition there are no hypotheses for RQ2. Age becomes an extra moderating variable. There are no hypothesis regarding age. The results show some minor differences between the 4 groups. In the abstact one can read: practices and identities play a vital role in moderating the degree of happiness. This seems a bit exagerrated. The answers where gathered with a face-to face interview. This could lead to social desirable answers regarding religion and political attitude, given the current political situation in China. For instance Lu and Goa found in China 22% religious identity and 36% religious practice. This investigation: identity: 10% and practice: 13.5%. What is meant with Christianity. Protestantism? Because Catholicism is also a branch of Christianity. Maybe a native English speaker could correct the text. Some examples:

-Pag2 , 65: index- indexes

-Pag2, 73: related= be related

-Pag 2, 83: context= contexts

-Pag 3, 101: scholar= scholars

-Pag 3, 106: question= survey?

-Pag 3, 136: is subjective= is a subjective

-Pag 3,  136: practice are = practices consist of

-Pag 3, 142: found pay= found to pay

-Pag 3, 141: survey= surveys

-Pag 5, 220, 22: grassroot= grassroots

-Pag 6, 239: question= questions

-Pag 6, 26: tend to be= tend to

-Pag 7, 292: samples= respondents

-Pag 7, 297: five Likert scales= a Likert scale

-Pag 7, 308: recode= recode the

-Pag 7, 310: analysis= analyze

-Pag 8, 339: the effect= the moderating effect’

-Pag 8, 363: samples = cases

-Pag 9, 390: of respondents= of the respondents

-Pag 9, 396: mediating= moderating

-Pag 9, 401: group RI*RP= 8%. I would also add the percentages of the other groups.

-Pag 14, 535: H3-1= H3-2.

-Pag 16, 621: mediators= moderators.

Author Response

Reviewer 3

Comment 1: This article focuses on religion as a moderator between determinant factors of happiness and happiness.  But I need some more argumentation why religion is treated as moderator and not as independent variable. There is no substantion for this choice from the literature. The argumentation for choosing religion as a moderator points at: a) there are mediators between religion and happiness en b) the frequency of participating in religious practices is low. Why is this a reason for choosing for moderation?

Authors’ response: Thanks for the reviewer’s questions. Of course, religion could be treated as independent variable, as plenty of researches have done before. However, as we explained in Line 136-167, in a highly secularized country like China and with low religious belief according to previous surveys, we try to propose another perspective to investigate the influence of religion by treating religion as moderator. And this idea did not come out of nowhere, the authors were inspired by the previous researches that had treated religious-items as moderators. Those researches have noticed the moderating effect of religion-related items and had valuable results in the western contexts, especially as moderators of indicators and subjective sense like well-being, so it’s reasonable to employ the current two religious variables as moderators, but not independent variables.

Comment 2: If one divides the moderating variable in 4 groups on the basis of religious participating and identity one expects an argumentation about the differences in these 4 groups regarding the connection between the determinants factors of happiness and happiness. But all the hypotheses are the same for the different groups. The hypotheses do no regard the moderating groups but the determinant factors. There are no hypotheses for group 4: no RI * no RP. RQ2 (is mental health a greater determinant of people’s happiness than physical health?) deviates from the purpose of the article. In addition there are no hypotheses for RQ2. Age becomes an extra moderating variable. There is no hypothesis regarding age.

Authors’ response: Thanks for the reviewer’s comments. We add hypothesis concerning age, which is QR4 (Line 277). QR 2 was deleted but we believe the finding about the differences between physical and mental health is valuable so we keep the findings.

About group 4 “no RI * no RP”,since the main focus about the paper is to investigate the moderating effect of religious identity and religious practice on the chosen indicators, the group without both the two moderating effects are not paid equal attention with the other three groups. But we still presented the general results of group 4.

Based on the previous researches, this paper directly employing religious identity and religious practice as moderators, so we do not rise hypothesis on this point.

Comment 3: The results show some minor differences between the 4 groups. In the abstract one can read: practices and identities play a vital role in moderating the degree of happiness. This seems a bit exaggerated. The answers where gathered with a face-to face interview. This could lead to socially desirable answers regarding religion and political attitude, given the current political situation in China. For instance, Lu and Goa found in China 22% religious identity and 36% religious practice. This investigation: identity: 10% and practice: 13.5%. What is meant with Christianity. Protestantism? Because Catholicism is also a branch of Christianity.

Authors’ response: Thanks for the reviewer’s questions.

It is true that the Chinese General Social Survey (CGSS) were conducted through face-to-face interviews every year. The main objective of the 2015 CGSS is to collect data on the general life issues of Chinese people, not specifically religion. Thus, the original investigators did not intentionally to get benefit on collecting information regarding religious issues. On the other hand, Lu and Gao (2017) used the data of 2007 Spiritual Life Study of Chinese Residents, which is a dataset specifically designed on religiosity in China, and collected through face-to-face interviews. Due to the difference on investigation aims, there might be different approaches when conducting interviews. Besides, the longitude CGSS data from 2003-2015 demonstrated a relatively stable results in same questions, including the religious ones. Hence, we would like to believe that the 2015 CGSS has exerted a minimum level on respondents’ religious issues, and the related data has an accepted quality standard.

About “基督教” in China, usually it means Protestantism,so thank you for pointing out, we will use “Protestantism” instead of “Christianity” in the paper for preciseness. Please be aware that the survey was conducted in Chinese, so the translation will not affect the research results.

References

Abdel-Khalek, A. M. (2014). Religiosity, health and happiness: Significant relations in adolescents from Qatar. International Journal of Social Psychiatry, 60(7), 656-661. doi:10.1177/0020764013511792

Abdel-Khalek, A. M., & Lester, D. (2017). The association between religiosity, generalized self-efficacy, mental health, and happiness in Arab college students. Personality and Individual Differences, 109, 12-16. doi:10.1016/j.paid.2016.12.010

Baroun, K. A. (2006). Relations among religiosity, health, happiness, and anxiety for Kuwaiti adolescents. Psychological Reports, 99(3), 717-722. doi:10.2466/pr0.99.3.717-722

Lu, J., & Gao, Q. (2017). Faith and Happiness in China: Roles of Religious Identity, Beliefs, and Practice. Social Indicators Research, 132(1), 273-290. doi:10.1007/s11205-016-1372-8

Sugiman, T. (2006). Group dynamics (in Japanese). In T. Sugiman (Ed.), Group dynamics in communities (pp. 19-86). Kyoto: Kyoto University Press.

Round 2

Reviewer 1 Report

After conducting two reviews of the paper I feel that the authors have made great efforts to address my comments and to improve the quality of the paper. I still feel that there are certain inherent weaknesses in the paper but on the whole the paper has been significantly improved in light of the comments made in my two previous reviews.

Author Response

Thank you very much for your comment this and your two previous reviews, we have improved the paper once again according to the suggestions.

Reviewer 3 Report

I agree with the revisions. Only one question remains. At page 3 the authors start a discussion with the following steps. 1. There is a correlation between religion and happiness. Religion is the independent variable. 2. This correlation can be different depending on the context. In poor contries this correlation exists, but not in rich contries. In this case the state of the country is the moderating variable. 3. The influence of religion on happiness is MEDIATED by other variables like autonomy, competence, relatedness. 4. In China participating in religious practices is low. This leads to the conclusion on page 4: may not support religion as a determinant of happiness but as a moderator. My question remains: what are the reasons related to the discussion on page 3 to expect moderating effects and which differences are expected between the 4 chosen groups of moderation? This because in the hypotheses no differences between the groups are expected.

Author Response

Thank you very much for your constructive suggestions. We have realized that some of the hypotheses might have misled readers in terms of the moderating function of religious belief and religious practice. Therefore, we have changed the proposal of our research hypotheses into five research questions respectively focusing on core points pertaining determinants and happiness. You may find the relating revisions in Line 163-166, 178-179, 197-198, 215-216.

 The reasons to use religion as moderator are revised and presented in Line31-33, 111-141. In simple terms, we employ religion as moderator based on the precious studies and the social context of China.